# MEIS1 in Hematopoiesis and Cancer. How MEIS1-PBX Interaction Can Be Used in Therapy

**DOI:** 10.3390/jdb9040044

**Published:** 2021-10-13

**Authors:** Francesco Blasi, Chiara Bruckmann

**Affiliations:** Foundation FIRC Institute of Molecular Oncology (IFOM), 20139 Milan, Italy; francesco.blasi@ifom.eu

**Keywords:** MEIS1, PBX1, MLL-r, leukemia, interaction surface

## Abstract

Recently MEIS1 emerged as a major determinant of the MLL-r leukemic phenotype. The latest and most efficient drugs effectively decrease the levels of MEIS1 in cancer cells. Together with an overview of the latest drugs developed to target MEIS1 in MLL-r leukemia, we review, in detail, the role of MEIS1 in embryonic and adult hematopoiesis and suggest how a more profound knowledge of MEIS1 biochemistry can be used to design potent and effective drugs against MLL-r leukemia. In addition, we present data showing that the interaction between MEIS1 and PBX1 can be blocked efficiently and might represent a new avenue in anti-MLL-r and anti-leukemic therapy.

## 1. Introduction

Homeobox transcription factor MEIS1 belongs to the three-amino acids loop extension (TALE) sub-family of homeobox proteins. MEIS1 was discovered because a retroviral integration close to its regulatory region induced murine myeloid leukemia [1]. Since then, the roles of MEIS1 in development, cell regulation, and transformation have been extensively investigated.

MEIS1 is highly expressed in the bone marrow, and its predominant and better-known role is in embryonic and adult hematopoiesis [2,3,4,5]. Along with *PBX1* and *HOXA9*, *MEIS1* is expressed in hematopoietic stem cells (HSCs) but downregulated during their differentiation [5].

MEIS1 regulates growth and differentiation during vertebrate development [6] and is widely expressed in the central nervous system and periphery, both during embryogenesis and after birth [2,3]. During embryo development, Meis1 ablation results in hematopoietic, vascular, and ocular defects [2,3]. It has been also well defined the role of Meis1 and Meis2 in mouse axial skeleton formation [7,8,9] and in anterior-posterior patterning for limb initiation [10], for which Meis loss-of-function (*Meis1* and *Meis2* double knock-out) embryos are limbless, with undeveloped appendicular condensations.

Genome-wide association studies have linked polymorphism of MEIS1 to an increased risk of “Restless legs syndrome” [11,12,13,14]. MEIS1 has also been linked to insomnia and sleep disorders [15,16,17]; in addition, it appears to be an essential regulator of the cardiomyocytes cell cycle [18,19,20].

A direct connection between MEIS1 expression, leukemia, and cancer has been established from the start (see below) [1]. The present review aims at describing the role of MEIS1 during embryonic and adult hematopoiesis, its implication in leukemia, and at establishing how MEIS1 direct targeting might become a valuable tool for the treatment of MEIS1-expressing malignancies, in particular during MLL-rearranged (MLL-r) leukemia.

## 2. PREP1 and MEIS1: Friend or Foe?

One of the properties of TALE proteins is to interact with each other. Previous studies have defined that the primary binding partner of MEIS1 is PBX (another TALE protein), of which exist four different genes [21,22]. MEIS1 and PBX form a stable heterodimer that cooperatively binds DNA [23,24]. In particular, the N-terminal domain of MEIS1, the MEINOX domain, also known as the Homology Region (HR), serves as the interaction surface with PBX [25,26,27]. There is an additional member of the TALE family, pKNOX1, better known as PREP1 (for PBX-regulating Regulating Protein 1), which also binds DNA as heterodimer with PBX [24,28].

PREP1 and MEIS1 share an overall similar organizational structure of the protein, in which the HR regions are almost identical. On the other hand, the homeodomain of MEIS1 and PREP1 is only partially conserved, which explains the slight distinction in DNA target specificity [29]. Conversely, the carboxy-terminal regions of the two proteins are dissimilar, justifying the functional differences [30]. Notably, since MEIS1 and PREP1 share the same HR domain to heterodimerize with PBX [31,32,33], they compete for this factor. Heterodimerization of MEIS1 or PREP1 with PBX is crucial for nuclear localization [31,34,35], resistance to nuclear exportins [36,37] and to provide PBX1 with the DNA specificity [29].

MEIS1-PBX1 and PREP1-PBX1 dimers can bind DNA and form trimeric DNA-binding complexes with HOX proteins. During embryonic stages, trimerization of MEIS1/PREP1 with PBX1 and HOXB1 is necessary to express several *HOX* cluster genes, playing crucial roles in development and hematopoiesis [5,38,39,40,41]. In human leukemia, the cooperation between MEIS1 and HOXA9 with PBX has been extensively reported [42,43,44,45,46]; specifically, the PBX3 form appears to be the prevalent form involved [47].

The interaction of MEIS1 and PREP1 with PBX may lead to overlapping functions as they both share their interaction and bind to DNA target sequences that are very similar, although not identical [29]. On the other hand, the two proteins may compete for PBX, and hence may have opposite functions. These possibilities have been verified: for example, in zebrafish, prep1.1 and meis1 have partially overlapping roles in *hox* genes regulation [41,48].

Likewise, the embryonic hematopoietic phenotypes of *Meis1^−/−^* and *Prep1^i/i^* mice are similar [49], but they do not reinforce each other: indeed, the lack of phenotype in double heterozygous (*Prep1^+/−^* -*Meis1^+/−^*) embryos was unexpected [29]. Therefore, even if both Prep1 and Meis1 affect embryonic hematopoiesis, they must act through different molecular pathways.

On the other hand, Meis1 and Prep1 compete in mouse tumorigenesis and leukemogenesis [32] because *Meis1* acts as an oncogene only in the absence of *Prep1* (or other tumor-suppressing genes), and the overexpression of Prep1 in Meis1-dependent tumor cells blocks their growth. In brief, Meis1 and Prep1 require Pbx in their respective tumorigenic or tumor suppressive functions and target the same molecular pathways, but in the opposite way [32].

The presence or absence of PREP1 has not yet been taken into account in the case of MLL-r, Mixed Lineage Leukemia, a leukemia due to translocation of several genes to the amino terminus of the *MLL* gene [50,51,52], which appears to be mainly dependent on the overexpression of MEIS1. In PBX-MEIS1-induced human leukemia and in MEIS1-dependent mouse tumorigenesis [47,53], HOXA9 appears to have an accelerating role [54].

## 3. MEIS1 during Hematopoietic Development

Meis1 is required during normal fetal murine hematopoiesis [2,3]. In mouse fetal HSCs, the expression of *Meis1* is tightly controlled. *Meis1* expression is stage-specific, with the highest levels observed in HSC and early progenitors; in later stages, *Meis1* expression decreases [5]. A homozygous knock-out of *Meis1* is embryonic lethal: knock-out embryos die by E-day 12.5–14.5 because of hemorrhages caused by the absence of megakaryocytes, platelets [55] and severe vascular defects [2,3].

In addition, there is a complete absence of functional HSC [56,57]. Additionally, knock-out mice for *Pbx1* die at the same embryonic stage and die presenting severe hematopoietic defects [58]. The degree of involvement of each partner is not well defined yet. *Prep1* deficiency appears to share a similar defect as *Meis1* in the fetal HSC. Indeed, in *Meis1* knock-out or *Prep1* hypomorphic (*Prep1^i/i^*) mouse embryos, fetal HSCs are deficient in number and regenerating ability and display various definitive hematopoietic defects [59,60,61]. In both cases, the observed phenotype appears to be due to a loss of cell cycle control in fetal HSC with a decrease of the number of cells in G0; this leads to an over replication of HSC, resulting in faster exhaustion [62].

Regulation by Meis1 and Prep1 is, therefore, crucial to preserve the self-renewal of the stem cells pool and the homeostasis of the hematopoietic system. In addition, this primary function of Meis1 in maintaining the self-renewal of the stem cells pool might be relevant in its oncogenic role. On this basis, one would expect that mice heterozygous for *Meis1* and *Prep1* should show an embryonal blood defect. However, this does not happen, indicating that the two transcription factors do not exploit the same mechanisms; this represents a still unsolved part of the physiology of Prep1 and Meis1 [29] and requires further investigation.

In zebrafish, meis1 shows over 90% amino acid sequence-identity with vertebrate orthologues and plays an essential role in embryonic hematopoiesis [63,64]. Anti-sense meis1-morpholino (MO-meis1)-injected embryos, where a MO complementary oligonucleotide blocks the translation initiation site of meis1 mRNA, lack mature erythrocytes. Since gata1 and alas2 hematopoietic markers are not affected, meis1 appears to exert its effect independently and to be required for terminal erythroid differentiation. While cell proliferation and cell death appear normal upon meis1 depletion, MO-meis1 injected embryos exhibit a reduced number of neutrophils and macrophages [63], witnessing a broader meis1 role in other hematopoietic lineages.

Furthermore, MO-meis1 display severe disruption of vessel lumen formation [63,65]: Cvejic and colleagues observed that 30 h post fertilization in MO-meis1, only a single vessel tube was formed, while in control embryos at the same stage, it was possible to distinguish dorsal aorta and posterior caudal vein forming two distinct tubes; therefore, the loss of meis1 impairs vessel lumen formation.

Cvejic and colleagues also tested a meis1 splice MO, designed to create aberrant splicing between exon 1 and 2. In spliced MO-injected embryos, exons 2 to 7 were removed, resulting in a meis1 truncated form, depleted of the pbx1-interaction motif and, therefore, incapable of binding pbx1. Similarly to the MO-meis1, meis1 splice MO-injected embryos displayed the same lack of red blood cells and reduced neutrophils and macrophages. Consistently, MO-pbx1 injected embryos showed a profound reduction in the number of erythrocytes and neutrophils and severe disruption of vascular patterning [63]. Therefore, the loss of meis1, pbx1, or of their interaction leads to similar phenotypes, suggesting that in zebrafish pbx1 is an essential binding partner of meis1 in hematopoiesis and vascular patterning.

## 4. Meis1 in Adult Hematopoiesis in Mouse

In a conditionally inactivated *Meis1* adult mouse hematopoietic system, as reviewed by Miller and colleagues [57] and summarized in Table 1, Meis1 appears to maintain the pool of primitive HSC. Meis1 depletion in HSC compromises cellular quiescence leading to stem cell exhaustion. In detail, the deletion of Meis1 causes a reduction of hematopoietic stem cells/progenitor cells (HSPCs) and, consequently, multi-lineage bone marrow failure [66]. Ariki and colleagues highlighted that Meis1 acts in replicating HSPCs rather than during their differentiation.

Deleting *Meis1* in all tissues of adult mice does not lead to hemorrhages, suggesting that the role of *Meis1* is essential in embryonic development but might be dispensable in the adult, especially for proliferation and differentiation of committed hematopoietic progenitors [57].

While, in the embryo, the mutation in *Prep1* leads to a major hematopoietic phenotype, similar to that of Meis1, adult conditional knock-out mice do not display a major hematopoietic defect. Moreover, the blood-regenerating activity of adult bone-marrow-derived *Prep1^i/i^* HSC is as efficient as wild-type HSC (Modica L. and Blasi F., unpublished data) [61]. Therefore, it appears that both *Meis1* and *Prep1* have different roles in the adult v. fetal hematopoiesis. However, the actual role of these two transcription factors in adult HSCs has not yet been carefully explored.

The effect of Meis1 knock-out is known to be partially mediated through reactive oxygen species (ROS). Indeed, loss of Meis1 leads to downregulation of hypoxia-inducible factors 1α and 2α (HIF-1α and Hif2a) and the consequent increase of ROS [68]. Moreover, an elevated ROS level in HSCs positively correlates with more defects in HSCs renewal and increased apoptosis [67]. It was already established that increased levels of reactive oxygen species (ROS) limit the lifespan of HSCs in vivo [69].

In this regard, significantly higher ROS levels were found in Meis1-depleted HSCs than controls, while the phenotype was rescued by the presence of the ROS scavenger N-acetylcysteine [68]. Therefore, when stem cells are under proliferative stress, the loss of Meis1 has a more significant impact, as, under these conditions, quiescent HSCs enter the cell cycle and exhaust their capacity to maintain hematopoiesis [68]. This effect observed under hypoxic conditions is possibly due to a downregulation of the hypoxia-response regulator Hif-1alpha [70], whose levels are regulated by Meis1 [68].

In conclusion, as summarized in Table 1, Meis1 ablation in adult mice does not prevent normal hematopoiesis; therefore, a partial inactivation (as expected by a specific drug) of *Meis1* should be well tolerated. Hence, MEIS1 might represent a valuable therapeutic target as one might hope to find few, if any, collateral effects in the hematopoietic system.

The many analogies of Meis1 and Prep1 roles in the hematopoietic system during the development and in the adult are remarkable. Therefore, given a possible therapeutic use of MEIS1 in human leukemias, it might be important to analyze better the common interactions with PBX.

## 5. The MEIS1-PBX1 and the PREP1-PBX Interaction Surface

To our knowledge, no attempt has yet been made to target the MEIS1-PBX1 interaction surface, which would appear an obvious therapeutic target because of the requirement of PBX for MEIS1 oncogenic activity. Since we have recently described experiments that have shed further light on the PREP1-PBX1 interaction site [71], and, since the amino acids sequence of the MEIS region interacting with PBX1 is (almost) identical to that of PREP1, we present now some, as yet unpublished, data that might stimulate research in that direction.

The large HOX/MEIS/PBX complex could offer several interfaces to be targeted to decrease their oncogenic function in leukemic patients. However, the development of small-molecule inhibitors of protein–protein interactions is challenging, especially for transcription factors, as surfaces are typically smooth and do not present the deep pockets present in enzyme active sites, making the development of small-molecule inhibitors more difficult [72,73,74,75]. This may be the case for the PREP1-PBX or the MEIS1-PBX interaction surface.

MEIS1 oncogenic function strictly requires heterodimerization with PBX [32,47,76]. Indeed, extensive genetic characterization has demonstrated that MLL-r dependent leukemic transformation in mice is dependent on the Meis1-Pbx interaction and not simply on Meis1 overexpression [76]. In addition, the specific switch-off of MEIS1 in MLL-r leukemia upon treatment with the menin/MLL interaction surface inhibitor VTP50469 [77] (see below) has demonstrated for the first time that HOXA9 downregulation may be dispensable for the regression of leukemia, i.e., that MEIS1 represents the ideal target.

Biochemically, PREP1 competes with MEIS1 for PBX1 binding, which agrees with the lower MEIS1 binding affinity to PBX1 (80 nM for MEIS1, 18 nM for PREP1 [71]. Two hydrophobic stretches within the HR1 and HR2 domains of PREP1 and MEIS1 contain the information necessary for dimerization with PBX in vitro and in vivo [32,47,71,78,79]: in PREP1, residues 52–80 (in the HR1) and residues 117–138 (in the HR2); in MEIS1, residues 54–100 (in the HR1) and residues 150–194 (in the HR2). These regions are very hydrophobic, leucine and isoleucine-rich, and positioned in such a way to suggest the formation of a leucine zipper.

Extensive scanning mutagenesis of the PREP1 HR1 and HR2 domains revealed that the residues most likely positioned in the “a” and “d” positions of the leucine zipper and, therefore, directly involved in the binding to PBX, are L63/L70 and L66/K73 in the HR1, and I122/ L129 and L125/L132 in the HR2 (Figure 1A,B) [71]. For MEIS1, a scanning mutagenesis analysis has not yet been performed, but considering the very high homology to PREP1, the MEIS1 residues directly involved in the leucine zipper can be estimated: they should be L83/I90 (in position “a”) and L86/L93 (in position “d”) for the HR1 and I149/L156 (in position “a”) and L152/L159 (in position “d”) for the HR2 (Figure 1C,D) [71].

Indeed, in PREP1 a quadruple mutation within either HR1 L63A/L66A/L67A/L70A, or HR2 I122A/L125A/L129A/L132A, is sufficient to abolish the interaction of PREP1 with PBX1 in a time-resolved fluorescence immunoassay (TR-FIA) and pull-down assays, and impairs the nuclear localization of PREP1 without altering the overall folding of the protein [71].

Like PREP1, MEIS1 requires dimerization with PBX to reach the nucleus. Therefore, we have cloned MEIS1-GFP in a retroviral vector and analyzed in mouse embryonic fibroblasts (MEFs) the ability of wild-type MEIS1 to reach the nucleus compared to quadruple mutants in the heptad repeats within HR1 or HR2 domains. In MEIS1, we mutated into alanine the potential hydrophobic residues corresponding to those identified in PREP1. Hence, in the HR1 of MEIS1 we mutated L83/L86/L87/I90 and in the HR2 L149/I152/L156/L159 (see Figure 1, for a comparison of PREP1 and MEIS1 heptad repeats). All MEIS1-GFP-tagged constructs were successfully expressed in MEFs as judged by flow-cytometry analysis (data shown in the Appendix A).

The subcellular localization of the transfected wild-type and mutants MEIS1-GFP was then determined by immunofluorescence (Figure 2A). Wild-type PREP1-GFP and MEIS1-GFP completely localize in the nucleus; as for PREP1, neither of the two MEIS1 quadruple mutants MEIS1-GFP L83A/L86A/L87A/I90A or MEIS1-GFP I149A/L152A/L156A/L159A can efficiently translocate to the nucleus; as it is shown in Figure 2A, actually, most of the mutant proteins localize in the cytoplasm. To express this in quantitative terms, we determined the ratio of the localization of MEIS1 (wild-type and mutants) between nucleus and cytoplasm, with respect to a nuclear marker and actin: the nuclei were marked with DAPI, and the whole shape of the cell by phalloidin.

The histogram of Figure 2B shows the ratio of expression of MEIS1 (wild-type and mutants) between nucleus and cytoplasm, highlighting a striking difference in the MEIS1 localization between wild-type and mutants. Finally, we confirmed that the MEIS1 quadruple mutants could not bind PBX using pull-down assays (Figure 2C): in fact, anti-GFP-beads coprecipitated PBX1 when the wild-type MEIS1-GFP was used but not in the case of the MEIS1 L83A/L86A/L87A/I90A mutant. The MEIS1 mutant I149A/L152A/L156A/L159A, pulled-down only 2% of PBX1 with respect to the wild-type (Figure 2C, left panel). The pull-down was normalized by blotting the same membrane with anti-GFP (Figure 2C, middle panel) and anti-vinculin (Figure 2C, right panel) antibodies. We conclude that both HR1 and HR2 domains of MEIS1 are sufficient and essential for MEIS1 dimerization with PBX1.

We previously reported that the inhibition of the PBX interaction can neutralize MEIS1 oncogenic properties [32,47,79]. It has also been already described that a Meis1 mutated in its HR2 domain could not cooperate with Hoxa9 in colony-forming assays or in in vivo leukemia assays [47]. Likely, this can be explained by the inability of the Meis1 mutant to access the nucleus, as we show for the MEIS1 I149A/L152A/L156A/L159A mutant in Figure 2. In addition, similar to PREP1 quadruple mutant in the HR1, the MEIS1 L83A/L86A/L87A/I90A mutant is unable to interact with PBX1, nor to access the nucleus (Figure 2A). The pull-down experiment, reported in Figure 2C, confirms that both MEIS1 L83A/L86A/L87A/I90A and L149/I152/L156/L159 mutants are unable to bind PBX1, as compared to wild-type MEIS1.

In mouse models, the critical Meis1 interactor tested has been Pbx1. However, in human patients, it appears that the critical isoform is PBX3. This should pose no problems because of the sequence identity of the PBX orthologues in this region. Indeed, Garcia-Cuellar and colleagues identified in PBX3 the stretch 78-PALFSVLCEIKEKTGLSIRGAQE-100 (Figure 1E) as fundamental for the interaction with MEIS1 [47]. More recently, systematic mutational analysis of all the hydrophobic residues of PBX1 has revealed that single mutations into alanine of residues I50, V80, L81 or I84 have a significant influence in TR-FIA in the binding to PREP1 [71].

Conversely, in TR-FIA, PREP1 and MEIS1 retain similar binding behavior to PBX1 upon the deletion of four PBX1 stretches: the deletion of residues 43–55 retains 60% PREP1 and MEIS1 binding; the deletion of residues 77–84 retains 30% PREP1 binding and 20% MEIS1; deletion of the PBX1 stretch 105–113 has no significant influence on binding to PREP1 nor MEIS1; and the deletion of 197–204 retains 50% PREP1 and 35% MEIS1 [71]. Consistently, the stretch that mainly affects the binding to both MEIS1 and PREP1 is 77-LFNVLCEI-84 (Figure 1F), which corresponds to the stretch identified by Garcia-Cuellar and colleagues in Pbx3 [47].

In conclusion, we believe these data suggest a possible use of drugs preventing the MEIS1-PBX1 interaction in leukemias, particularly in MLL-r (see below). An even more accurate biochemical characterization of the MEIS1-PBX interface will further narrow the region to target and aid in developing small molecules effectively and specifically disrupting the MEIS1-PBX interaction.

## 6. MEIS1 Overexpression in Hematopoietic Precursors

Many studies in the past have detailed the contribution of MEIS1 in cell transformation. Primary wild-type Mouse Embryonic Fibroblasts (MEFs) are never transformed by a single oncogene, like Ras. Neoplastic oncogene-induced transformation instead occurs when at least two oncogenes are used or when a tumor suppressor gene is also missing [80]. Indeed, co-expression of MEIS1 with HoxA9 can induce growth factor-dependent oligoclonal acute myeloid leukemia in less than three months in mice [54,81].

Furthermore, at the genetic level, the cooperation of Meis1 and Hoxa9 enables the transcription of Flt3, which is not efficiently activated by Hoxa9 alone [82]. More recent data revealed that Meis1 alone could transform MEFs, but only in the absence of a tumor suppressor like p53 or Prep1 [32,53,83]. This data reinforced the role of Prep1 as tumor-suppressor and of Meis1 as oncogene. Indeed, in *Prep1^i/i^* MEFs, a second oncogene (for example, HoxA9) is, in fact, not required to induce transformation by MEIS1 overexpression.

The neoplastic transformation was demonstrated with all classical assays, including colony formation in soft agar and the injection of transformed cells in mice [32]. Using the same assays, this cellular system allowed to test the competition between Meis1 and Prep1 and delineate a possible mechanism. The growth in mice of *Prep1^i/i^* MEFs overexpressing Meis1, as well as soft agar colony formation, was drastically reduced when Prep1 was re-introduced. This result allowed the testing of Prep1 mutants that had lost part of their domains.

While the loss of the C-terminal fragment or the DNA-binding homeobox did still inhibit the growth of the Meis1 overexpressing cells in mice, the loss of the amino-terminal HR1 and HR2 abolished the ability of Prep1 to compete. As this domain is strongly and uniquely involved in the interaction with Pbx [31,71,84], this result strongly suggests that the oncogenic activity of Meis1 as well as the competing activity of Prep1, both require the interaction with Pbx proteins [32].

RNA-seq data showed that the type of target pathways hit by both transcription factors were identical in the two cell types, except that Meis1 hits the promoting genes, while Prep1 hit the inhibiting ones [83]. The ability of Meis1 to interact with Pbx proteins is essential also for the induction and maintenance of MLL-mediated myeloid transformation [76]. In this system, both the deletion of the HR1+2 domain of Meis1 and the depletion of Pbx1 abolish Meis1 oncogenic activity [79].

Thus, both Meis1 oncogenic and Prep1 tumor-suppressing activities require interaction with Pbx1. Indeed, the Pbx-interacting regions (HR1 and HR2 domains) [31,34,71] are required in both the oncogenic function of Meis1 and in the tumor-suppressive role of Prep1, i.e., both activities are performed by a complex of either protein with Pbx1. Therefore, Pbx1 can have a dual role as an oncogene or tumor suppressor, depending on its binding partner (Mesi1 or Prep1, respectively).

## 7. The Role of MEIS1 in Leukemia, in Particular in Mixed Lineage Leukemia (MLL-r)

Acute leukemia arises when hematopoietic progenitor cells become locked into an immature state of perpetual proliferation. As HOXA9 and MEIS1 are involved in maintaining the HSC pool, aberrant overexpression of HOXA9 and MEIS1 can drive leukemogenesis [45,85].

By using quantitative real-time PCR to investigate gene expression in 100 children with acute leukemia and comparing them to those of healthy controls, Adamaki and co-workers [86] showed that aberrantly high HOXA9 and MEIS1 gene expression was associated to different leukemic subtypes, including various maturation stages of B-cell ALL and other cytogenetic MLL-positive populations, suggesting that these genes are involved in the development of a broad range of leukemic subtypes in childhood.

The overexpression of MEIS1 is implicated in childhood leukemia (AML and ALL), including B-cell ALL [86]. Thus, high HOXA9 and MEIS1 expression have indeed the potential to be used as additional prognostic markers of relapse and overall survival of acute childhood leukemia at the time of diagnosis. In addition, AML patients with high expression of MEIS1 had a shorter survival time than those exhibiting low MEIS1 expression, and high MEIS1 level was associated with an inadequate response to conventional chemotherapy [87].

MLL-r is a particular type of leukemia. The chromosomal translocation of the mixed-lineage leukemia 1 gene (MLL1, also known as MLL or KMT2A) accounts for ~9% of adult, ~5% of children, and 80% of infants cases in acute lymphoblastic leukemia (ALL), together with over 10% of adult cases, ~15% of children, and above 70% of infant cases in acute myeloid leukemia (AML) [88,89].

The MLL1 fusion proteins (MLL-AF9, MLL-AF4, and MLL-ENL) lack the SET domain necessary for catalyzing the mono- di and trimethylation of histone H3 lysine 4 (H3K4) since the methyltransferase activity of MLL1 is compromised. MEIS1 and HOXA9 have been identified as the most critical targets of MLL fusion proteins in MLL-r [4,45,82,86,90,91,92,93,94]. It is well established that HOXA9 and MEIS1 overexpression drives leukemogenesis in MLL-r leukemia [95]. However, recent data question the driving role of HOXA9 (see below).

MLL-r leukemia has been extensively studied in the mouse by the laboratory of Michael Cleary, which has established the essential role of Meis1 in leukemia induction and maintenance. In any case, Meis1 does not act on its own but, as in its all-other known functions, as a dimer of one of the Pbx proteins. In mammals, there are four *Pbx* genes, of which *Pbx1* and *Pbx3* express two different forms, short and long. In human MLL-r, the form of PBX that appears to be mostly involved is PBX3. However, all data available indicate that PBX1 and PBX3 are indistinguishable in terms of MEIS1 binding. From this point of view the dimer MEIS1-PBX3, and not just MEIS1, appears to be an interesting target for therapy.

Liu and colleagues [96] identified a distinct expression pattern of MEIS1 in a cohort of 95 patients with newly diagnosed AML without MLL abnormality. In addition, MEIS1overexpression is involved not only in patients that carry MLL-r AML but also in patients that have nucleophosmin (NPM1)-mutant AML. NPM1 is often mutated in AML by a frameshift that causes loss of its C-terminal nucleolar localizing segment. Consequently, mutated NPM1 localizes exclusively in the cytoplasm (NPM1c), which perturbs several cellular functions and supports leukemogenesis.

AML harboring NPM1 mutations represent one of the most common genetic lesions in adult AML. Moreover, it has a similar stem cell-associated gene expression program, which includes a well-characterized deregulated expression of the HOXA cluster genes and MEIS1 [45,97,98,99,100,101,102,103]. In addition, using a CRISPR/cas9 negative selection screen, Kühn and Armstrong confirmed that the binding of wild-type MLL to menin was critical for the expression of HOXA9, MEIS1, and FLT3 in NPM1-mutated AML cells [104]. This evidence also suggests that the expression level of MEIS1 might be a prognostic factor for AML patients even without the expression of other fusion genes.

## 8. The Role of MEIS1 in Solid Cancers

MEIS1 also plays a role in solid tumors; however, in this case the MEIS1 role is contradictory as it may behave as an oncogene or a tumor suppressor. The role of the whole MEIS protein family (MEIS1, MEIS2, and MEIS3) in solid cancers has been recently reviewed by Girgin and colleagues [105]. MEIS1 overexpression in several solid cancers has an important oncogenic function, for example, in breast [106,107], colorectal [108], human esophageal squamous cell [109,110], neuroblastoma [111], ovarian [112], and prostate cancers [113], where its upregulation has been associated to cancer etiology, progression, and increased invasiveness.

For example, Zargari and colleagues evaluated the expression pattern of MEIS1 in human esophageal squamous cell carcinoma, demonstrating that down-regulated expression of MEIS1 by shRNA decreased the mRNA expression of the most important stem cell markers (such as BMI1, SALL4, OCT4, and KLF4), which can preserve self-renewal and proliferative potential via inhibiting differentiation signaling pathways during cancer initiation and development [114]. However, it has been recently demonstrated that there is an inverse relationship between MYC activity in prostatic cancer, which drives prostate cancer progression, and MEIS1 expression.

Upregulation of MYC decreases MYC occupancy at the MEIS1 locus and leads to downregulation of MEIS1. On the other hand, knockdown of MYC activity leads to increased expression of MEIS1, following an increased occupancy of MYC at the MEIS1 locus [115]. In addition, in prostatic cancer tissues, statistically significant positive correlations between MYC and HOXB13 mRNA levels was observed [115], while an independent group confirmed that the tumor-suppressive activity of MEIS1 was dependent on HOXB13 [116].

In addition, MEIS1 seems to function as a tumor suppressor in the progression of clear-cell renal cell carcinoma [117], and similarly in lung cancer [118]. In conclusion, while the oncogenic role of Meis1 in hematological malignancies seems more than likely, the same cannot be asserted in solid cancers.

As described above, in mouse embryonic fibroblasts, Meis1 overexpression can directly transform in the absence of additional oncogenes, provided a tumor suppressor gene is absent [32]. In this system, the overexpression of Prep1 can inhibit tumorigenesis. These functions, both requiring the interaction with Pbx, are associated with a curious behavior of these two proteins when overexpressed. Indeed, ChIPseq data show that, while physiologically expressed Prep1 or Meis1 bind to their specific target sequences, overexpressed Prep1 or Meis1 tend to mainly target genes that are normally bound by the AP1 oncogene (one of the fos-jun dimers), which has just a similar DNA target recognition.

Although this data has not been further explored, this spurious binding has a counterpart in the gene ontology of the genes bound. Indeed, overexpression targets both Prep1 and Meis1 towards the cell growth pathways, with a significant difference. Although the pathways are the same, Prep1 binds to genes inhibiting growth, whereas Meis1 binds to genes activating growth [83]. It would be interesting to test whether this difference is due to the different carboxy termini of the two proteins.

## 9. MEIS1 and HOXA Inhibitors in Cancer Therapy

### 9.1. Drugs Indirectly Targeting MEIS1, though Epigenetic Control of Its Expression

Overexpression of the HOX and MEIS1 genes triggers leukemogenesis and is associated with high-risk AML but up to now could not be directly targeted by drugs. Several epigenetic control strategies to decrease HOXA9 expression have been developed in the last decade. Currently, it is well recognized that HoxA9 expression alone is only weakly oncogenic in mouse leukemia models and usually requires a second “hit”, the overexpression of Meis1 [54,119].

As described above, recent results have highlighted a crucial role of MEIS1 in maintaining AML, while the role of HOXA9 is dispensable. Indeed, MEIS1 functions is essential during the progression of human AML and is an adverse prognostic factor independent of MLL1 abnormality [96]. Thus, MEIS1 is a critical oncogenic factor in the development of acute leukemias, and its inhibition represents a promising strategy to reverse its oncogenic activity.

Some epigenetic modulators, even of factors deregulated in leukemia, are used in clinics or have entered clinical trials. Lambert and colleagues [49] summarized in great detail all the strategies previously undertaken to target MLL-interacting partners at the protein–protein interaction level to modulate HOXA9 levels in MLL-r. Notably, most of these drugs downregulate the levels of MEIS1. More recently, Yao and colleagues [120] reported the current status on the development of MEIS1 inhibitors as therapeutic agents.

#### 9.1.1. Targeting the Enzyme DOT1L (KMT4), Which Methylates Lysine 79 of Histone H3 (H3K79)

DOT1L methyltransferase activity is critical to MLL-r leukemia [121] and is recruited on DNA where it induces hyperexpression of HOXA9 and MEIS1 [122]. During the past few years, inhibitors of DOT1L have shown potent and selective activity against MLL1-r leukemia [123,124,125]. Indeed, it was recently reported that significant MEIS1 down-regulation is a typical result of the DOT1L inhibition [126].

DOT1L inhibitors, such as EPZ004777 and EPZ5676 (also known as pinometostat), reduce HOXA9 and MEIS1 mRNA level expression and induce differentiation of AML cells [127]. In preclinical testing, pinometostat inhibited the proliferation of leukemia cell lines harboring MLL-r and induced sustained regressions in MLL-r rat xenograft models. However, clinical responses were modest in treated adult patients, inducing remission only in 2 of 51 patients. Therefore, DOT1L inhibition attained by pinometostat as a stand-alone therapy is not sufficient to achieve clinical benefit in patients with relapsed/refractory MLL-r leukemia [128].

Another selective inhibitor of DOT1L, SYC-522, an S-adenosyl-L-methionine (SAM) derivative, blocks the cell cycle at the G0/G1 phase and sensitizes MLL-r leukemia cells to chemotherapeutic agents (mitoxantrone, etoposide, and cytarabine), causing apoptosis. After 3–6 days of treatment with SYC-522, the expression levels of HOXA9 and MEIS1 also decreased in AML cells by 50% or more. Decreasing HOXA9 and MEIS1 gene expression, SYC-522 promoted cell differentiation [129]. Therefore, inhibition of DOT1L by SYC-522 treatment shows that reducing DOT1L methyltransferase activity and its downstream targets MEIS1 and HOXA9, can promote differentiation and might represent a valuable approach for treating MLL-rearranged leukemia.

#### 9.1.2. Targeting the Menin/MLL Interaction Surface

Methylation of the histone-H3 lysine-4 (H3K4) by the methyltransferase KMT2A (aka MLL1) is an epigenetic mark associated with actively transcribed genes and a master regulator for the transcription of genes important for embryonic development and hematopoiesis (including HOX and MEIS1 genes), but dispensable in the mature cell. MLL1 is frequently upregulated in cancers, resulting in increased expression levels of HOX and MEIS1 target genes, which link MLL1 with its tumorigenic properties [45,94,130].

In MLL-r leukemia, chromosome translocation produces an oncogenic fusion protein consisting of the N-terminal DNA-interacting domains of MLL1 fused with one among over 60 fusion partners [52,131]. The translocation partner is canonically a transcription factor that becomes functionally deregulated and promotes the typical leukemic target gene expression pattern, which includes the overexpression of FLT3, HOXA9, and MEIS1. Transcription of the latter two genes promotes self-renewal and is critically dependent on the MLL1-fusion protein binding to menin. Menin is, therefore, another druggable component of the MLL aberrant complex in MLL-r leukemia.

Inhibition of the MLL-menin interaction in MLL-r cells leads to downregulation of MEIS1 and HOXA9 and other MLL-fusion target genes. Most importantly, the disruption of the MLL-menin complex abrogates the oncogenic properties of MLL fusion proteins [132].

MI-1, MI-2, MI-3, and MI-NC [93,133] represent the first generation of menin-MLL inhibitors. These inhibitors were able for the first time to pharmacologically inhibit the MLL-menin complex formation in vitro and consequently to substantially decrease the expression levels of HOXA9, MEIS1, and other genes dependent on the aberrant MLL-menin interaction. MI-2, in particular, competes with MLL for menin binding by targeting menin MLL-binding pockets and 12 µM MI-2 effectively disrupts cellular menin and MLL-AF9 fusion interaction by >90% in MLL-AF9-expressing HEK293 cells.

Furthermore, MI-2 was shown to inhibit the oncogenic proliferation and to induce apoptosis of four MLL translocation-harboring human leukemia cell lines (by >95% in 12 d at 6 µM), but not of non-MLL leukemia lines Kasumi-1 and HAL-01 (≤16% inhibition at 12 µM) [133]. At the same time, they downregulated MLL target genes expression via simultaneous induction of hematopoietic differentiation and killing of non-differentiated population. MI-2 reduced by 80% HOXA9 and MEIS1 expression, induced differentiation, and had anti-clonogenic activity in methylcellulose assays [133]. However, MI-1, MI-2, MI-3, and MI-NC were unsuitable for in vivo studies because of moderate cellular activity and poor pharmacological properties.

Other reversible and irreversible menin inhibitors have been developed in the last years, all suppressing MEIS1 and HOXA9 expression without affecting adult hematopoiesis: MI-463 and MI-503 [134], MI-3454 [135], and M808 [130]. They all represent potential new therapies for the treatment of MLL leukemia, as they have demonstrated modulation of the expression of HOXA9 and MEIS1 in human MLL-r leukemic cell lines (MV-4-11 or MOLM13) and a substantial survival benefit in MLL-r leukemia model mice, through the promotion of hematopoietic cell lineages differentiation, reaching complete remission of leukemia. Furthermore, these inhibitors display pronounced growth suppressive activity in MLL1-r or NPM1-mutated leukemia mouse models and a minimal effect in human leukemia cell lines without MLL translocations.

Krivtsov and colleagues [77] recently characterized VTP50469 (a menin-MLL interaction small-molecule inhibitor), which presents potent anti-leukemic activity, and identified novel downstream gene regulation. The authors have modelled, in silico, the VTP50469 molecule, taking advantage of the available crystal structure of the MLL-menin inhibitor, MI-2, bound in the MLL binding pocket of menin [93]. VTP50469 displaces menin from the protein complex and inhibits the chromatin occupancy of MLL at selected genes.

Loss of MLL binding led to changes in gene expression, differentiation, and induced apoptosis. Interestingly, VTP50469 antiproliferative activity in MLL-r cell lines induced the repression of MEIS1 but left the HOX-gene transcription intact and resulted in low toxicity. Inhibitor VTP50469 demonstrated for the first time that effective targeting of only MEIS1 is sufficient to abrogate the oncogenic properties of MLL-r cells. Mice treated with VTP50469 had substantial reductions of cancerous tissue, and many animals remained disease-free several weeks after treatment. Further research involving patient-derived xenograft models of NPM1-mutated AML also demonstrated responses to the menin inhibitor VTP50469 [136], again through an effective downregulation of MEIS1 expression.

To date, two menin-MLL1 inhibitors (K0539, NCT04067336 and SNDX-5613, NCT04065399) have demonstrated preclinical activity in HOXA/MEIS1 genes-driven leukemias, including MLL-r and NPM1-mutant AML, and have entered clinical testing with a focus on patients with MLL-rearrangements or NPM1 mutations.

#### 9.1.3. Targeting WDR5/MLL Interaction through an MLL1 Peptidomimetic

The catalytic activity of MLL1 is regulated by the formation of a multi-subunit complex consisting of MLL1, WD repeat domain 5 (WDR5), retinoblastoma binding protein 5 (RbBP5), and the Absent Small Homeotic-2-like protein (Ash2L) [137]. In particular, the interaction between WDR5 and MLL1 plays a crucial role in regulating the H3K4 methyltransferase activity of MLL1 in physiological and pathological conditions [138]. The co-crystal structures of an MLL1 peptide complexed with WDR5 obtained by two groups [139,140] show that the interaction between WDR5 and MLL1 involves a distinct pocket in WDR5 and an MLL1 WDR5-interacting (WIN) motif comprised of approximately 12 amino acid residues.

Starting from the 12-mer MLL1 WIN peptide and through systematic analysis, Karatas and colleagues [141] determined the minimal binding motif in the MLL1 peptides required for the high binding affinity to WDR5. This resulted in the design of MM-102, an MLL peptidomimetic tri-peptide that binds to WDR5 with Ki ≤ 1 nM [142]. Evaluation of MM-102, in bone marrow cells transduced with MLL1-AF9 fusion construct showed that the compound decreases both HOXA9 and MEIS1 expression in leukemic cells expressing the MLL1-AF9 fusion gene [142]. MM-102 appears effective in inhibiting cell growth and inducing apoptosis in leukemic cells harboring MLL1 fusion proteins [142]. Although effective in cellular models, peptide-based inhibitors have shown poor cell permeability and metabolic stability in vivo.

#### 9.1.4. Spleen Tyrosine Kinase (SYK) Inhibitor ENTO (Entospletinib)

Spleen tyrosine kinase (SYK) signaling is another proposed target in AML. SYK promotes cellular differentiation and survival, and is expressed broadly in most hematopoietic cells. SYK expression is modulated by HOXA9 and MEIS1 [92]. More specifically, SYK protein levels, but not mRNA levels, are upregulated upon overexpression of MESI1 through a post-transcriptional mechanism [92]. Upregulation and activation of Syk by Meis1 has been identified as a key regulatory mechanism in a *Hoxa9/Meis1*-driven tumor in a mouse model.

Inhibition of Syk disrupts the *Hox9*/*Meis1* regulatory loop and prolongs the survival of mice. Sensitivity to Syk inhibition has been linked to HoxA9 and Meis1 overexpression in preclinical studies [143]. Entospletinib (GS-9973) is a small-molecule SYK inhibitor that disrupts the kinase activity of the enzyme, showing four-fold higher selectivity for SYK than for other kinases (such as JAK2, c-KIT, FLT3, RET, and KDR). Notably, patients with increased HOXA9 and MEIS1 expression had improved survival upon treatment with entospletinib; this indicates that HOXA9 and MEIS1 overexpression might be dependent on SYK signaling.

Entospletinib has been shown to produce complete remission rates of 83–90% in a phase Ib/II trial (NCT02343939) when administered in combination with chemotherapy to patients with AML bearing a HOXA9/MEIS1 signature (including those with FLT3-ITDs, NPM1 mutations or MLL1-r) [143,144], suggesting that for AML patients with increased HOXA9 and MEIS1 expression, SYK inhibition might represent a potentially valuable therapy.

### 9.2. Drugs Directly Targeting the MEIS1-DNA Interaction

Direct MEIS1 downregulation in the treatment of AML represents an attractive target [100]. However, changing epigenetic marks might not always be feasible in AML leukemic cells, as such approaches might present side and off-target effects altering normal cells.

Based on the high-resolution structure of the MEIS1 homeodomain, Turan and colleagues [145] recently developed MEIS-specific inhibitors by targeting the MEIS homeodomain and impairing MEIS ability to bind to its DNA target motif TGACAG in vitro. Turan and colleagues identified the residues F326, W327 R331 and R333 that are highly conserved and interact with DNA and performed an in-silico screening with small molecules based on the DNA-binding, to eliminate small molecules not specifically binding crucial residues.

The two inhibitors identified using high-throughput in silico screening (MEISi-1 and MEISi-2) effectively reduced MEIS-Luc reporter activity in a dose-dependent manner. Furthermore, MEISi-1 and MEISi-2 were also effective in downregulating HIF-1a and Hif-2a in hematopoietic cells and several HSC quiescence modulators. At the same time, the expression profile of other TALE proteins, such as PBX, HOXA9, TGIF, and PREP1, were unaltered in vivo.

Turan and colleagues demonstrated that both MEISi-1 and MEISi-2 injected intraperitoneally into wild-type mice were able to downregulate MEIS expression profile, induce HSCs expansion in the bone marrow (c-Kit+, Sca1+, CD150+cells, LSK HSPCs, LSKCD34low cells, and LSKCD150+CD48-cells), and downregulate MEIS target genes. The study of Turan and colleagues highlights the possibility to use MEIS1 DNA binding inhibitors as therapeutics for directly modulating the activity and the proliferation of HSCs.

## 10. MEIS1 Stability

One of the distinctive properties of the Meis1 and Prep1 dimerization with Pbx is often their increased stability in vivo [32,42,47]. When Meis1 is not in complex with Pbx it is destabilized and addressed to proteasomal degradation [47]. A similar effect is observed with Prep1. The half-life of Prep1 in the absence of Pbx is decreased, although not in all cells [146]. This is true also in the competition of Meis1 and Prep1 for Pbx [32]: in transformed fibroblasts (*Prep1^i/i^*, overexpressing Meis1), Meis1 acts as an oncogene and Prep1 antagonizes Meis1: when Prep1 is present, the oncogenic phenotype is suppressed/inhibited [32], and is accompanied by a decrease of Meis1. Indeed, as PREP1 affinity for PBX1 is five-fold greater than MEIS1 [71], it can effectively compete with MEIS1 for PBX1.

In addition, Garcia-Cuellar and colleagues demonstrated that a 5-alanine mutation in the HR2 (L152A/F154A/L156A/L157A/L159A) partially stabilized Meis1 by prolonging the protein half-life and blocking proteasomal degradation. Indeed, in a cellular system expressing the Meis1-Pbx3 complex, Meis1 wild-type, or Meis1 mutant in the presence of cycloheximide decreased, after 2 h of treatment, the amount of Meis1-Pbx3 complex to 60%. At the same time, the amount of mutant Meis1 decreased to 20% and that of wild-type Meis1 to almost zero.

The instability of Meis1 is due to its proteasomal degradation that leads to a short half-life when not in complex with Pbx3; complex formation with Pbx3 completely prevented the generation of ubiquitin-modified Meis1. Possibly, Pbx3 competes for the binding of a ubiquitin ligase; additionally, in immunoprecipitation studies, it was demonstrated by the same authors that in vitro Pbx3 increases Meis1 affinity for Hoxa9 in the presence of DNA [47]. As mentioned above, PBX3 (not PBX1 or PBX2) is apparently the most important form of PBX that interacts with MEIS1 in human MLL-r leukemias [42,47].

## 11. Future Perspectives and Challenges: Targeting MEIS1/PBX Interface for AML

As most potential Meis1 inhibitors thus far developed are intended to act in the nucleus, they have to overcome the nuclear membrane barrier. It might be, therefore, more convenient to target MEIS1 before it reaches the nucleus, i.e., while or immediately after it has been synthesized. Newly synthesized MEIS1 or PREP1 immediately dimerizes with PBX, forming a stable complex that is then able to enter the nucleus and, in addition, acquire the ability to bind DNA.

Therefore, preventing the formation of a MEIS1-PBX dimer would have a dual effect: not only the prevention of DNA binding before but also not allowing the localization of MEIS1-PBX to the nucleus. The MEIS1-PBX interaction domain was identified a long time ago, but only recently, due to its homology to the PREP1-PBX1 interaction domain, were the essential residues identified in detail [71]. In addition, not only inhibitors of MEIS1 DNA binding but also the inhibitors of menin and MLL must act in the nucleus; therefore, they also need to breach the nuclear membrane efficiently.

Transcription factor MEIS1 drives myeloid leukemogenesis in the context of HOX gene overexpression; however, MEIS1 is still considered a challenging transcription factor to target, for the absence of structural details, besides the homeodomain [147]. Nevertheless, the direct targeting of MEIS1 might provide further avenues for inhibiting the MEIS1/HOX-mediated leukemic transcription program. For example, the recent detailed knowledge of the molecular interaction between TALE oncogenic transcription factors [71] allows the design of novel therapeutics, such as small peptide inhibitors.

## Figures and Tables

**Figure 1 jdb-09-00044-f001:**
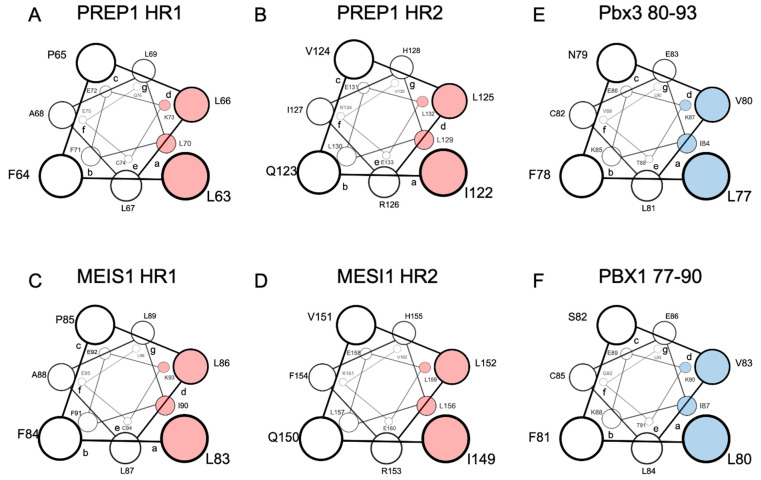
Schematic representation in a top view perspective of the alpha-helices of PREP1 and MEIS1 involved in the leucine zipper with PBX1. For MEIS1 and PREP1, the residues in positions “a” and “d” of the alpha helix, crucial for the interaction with PBX1, are highlighted in red for Pbx3 and PBX1; the important residues for binding MEIS1 and PREP1 are highlighted in blue. Panel (**A**): PREP1 HR1; panel (**B**): PREP1 HR2; panel (**C**): Meis1 HR1; panel (**D**): Meis1 HR2; panel (**E**): Pbx3 80-93; panel (**F**): PBX1 77-90. HR stands for “Homology Region”.

**Figure 2 jdb-09-00044-f002:**
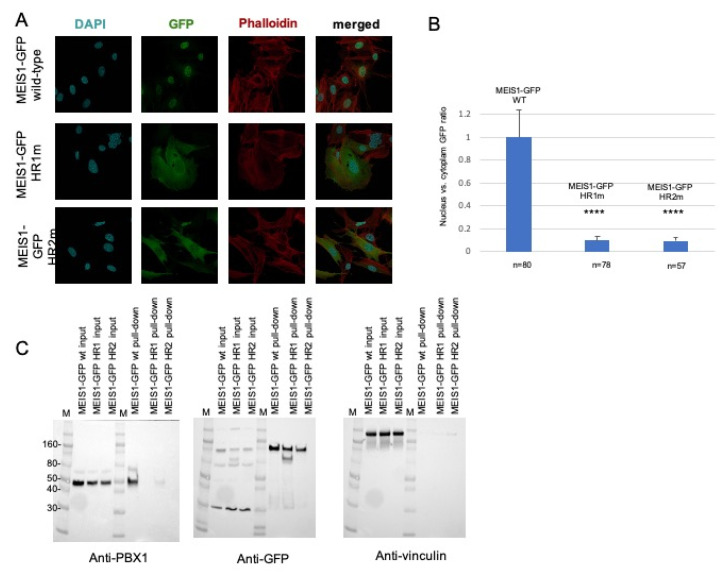
Mouse embryonic fibroblasts were transfected with a plasmid encoding wild-type or mutant MEIS1-GFP (HR1m is the MEIS1-GFP mutant L83A/L86A/L87A/I90A; HR2m is the MEIS1-GFP mutant I149A/L152A/L156A/L159A). Panel (**A**): The figure shows illustrative confocal immunofluorescence microscopy images in which the nucleus is identified by DAPI fluorescence and MEIS1 by GFP immunofluorescence. The transfected constructs are indicated at the left of the images. MEIS1-GFP is in green, DAPI in blue, and phalloidin in red. The merged DAPI, red, and GFP channels are shown in the right panels. MEIS1-GFP HR1 and HR2 mutants reveal a cytoplasmic localization, compared to the exclusively nuclear localization of wild-type MEIS1-GFP. Panel (**B**): Bar graphs showing the intracellular distribution of wild-type and mutants MEIS1-GFP. Plotted is the percentage of the total signal in each compartment, using measurements from between 57 and 80 individual cells for each construct. Only cells expressing MEIS1-GFP were included in the total counts and used to calculate percentages. Panel (**C**): pull-down experiment from total lysates. On the left, blotted with anti-PBX1 antibody, the inputs of the three total lysates of MEFs overexpressing wild-type MEIS1-GFP, and the two HR1 and HR2 mutants, along with the pull-down result. In the middle, the same membrane, blotted with anti-GFP antibody, to control the efficiency of the pull-down; on the right, the same membrane blotted with anti-vinculin, to control the loading of the input. M is the molecular weight standards, and some reference markers are annotated on the left side of the membrane. We indicate with **** values below 0,005 (99% significance).

**Table 1 jdb-09-00044-t001:** Summary of the effects of MEIS1 deletion in the hematopoietic cells of adult mice.

Effect of Meis1 Knock-Down in the Adult Mice	Conditional KO	Reference
Reduction in GMPs, CMPs, and MEPs Reduction in CFU-GEMM No changes in CFU-GM or BFU-E No changes in T, B, myeloid, and erythroid lineages in BM and peripheral blood	Deletion of exon8 on *Meis1* gene Tamoxifen injected *Meis1* fl/fl Scl-Cre-ER^T^ mice Ablation of Meis1 in long-term-HSCs	[67]
Decline in LT-HSCs Reduction of hematopoietic cells: B-cells. T-cells, monocytes, granulocytes Mild decrease of platelet count Progenitors unchanged Higher levels of ROS	Deletion of exon8 on *Meis1* gene Tamoxifen injected *Meis1*- fl/fl Rosa26-Cre- ER^T^ mice Ablation of Meis1 in long-term- HSCs	[68]
Reduction in the total number of BM cells Decrease of hematopoietic precursors CFU-Megakaryocytes Loss of quiescent HSCs Loss of HPCs	Deletion of exon8 on *Meis1* gene poly(I:C) injected *Meis1* fl/fl Mx1-Cre-ER mice Ablation of Meis1 in HSC/HPC	[66]

List of abbreviations used in the table: HSC (Hematopoietic Stem Cell); HPC (Hematopoietic Progenitor Cell); GMP (granulocyte-monocyte progenitor); CMP (common myeloid progenitors); MEP (megakaryocyte-erythrocyte progenitor); CFU (colony-forming unit); CFU-GM (Granulocyte, Monocyte); CFU-GEMM (granulocyte, erythrocyte, monocyte, megakaryocyte); BFU-E (Burst Forming Units-Erythroid); and ROS (Reactive Oxygen Species).

## Data Availability

Not applicable.

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
