# Peer review of "MEIS1 in Hematopoiesis and Cancer. How MEIS1-PBX Interaction Can Be Used in Therapy"

_jdb, 2021, doi:10.3390/jdb9040044_

Round 1

Reviewer 1 Report

In this review, the authors focus on the role of the TALE-HD protein MEIS1 in hematopoiesis and cancer. The topic is of interest and well documented, with an exhaustive description of the literature implicating MEIS1 in MLL-rearranged (MLL-r) leukemia. In addition, the authors describe in detail various published data showing how indirect and direct inhibition of MEIS1 activity can reduce cancer cell proliferation. They convincingly propose a novel potential strategy to block MEIS1 activity through the design of small molecules that would block MEIS1/PBX1 interaction, a mechanism allowing MEIS1 translocation into the nucleus and thus its ability to regulate gene expression and cell proliferation. In support of this strategy, they show experimental data suggesting that indeed mutations in the PBX1 interacting domains affect the subcellular distribution of a MEIS1::GFP fusion protein. Overall, the review is well written and organized in specific sections, each addressing a particular question. The text would benefit from some corrections, especially in sentences like “The histone-H3 lysine-4 methyltransferase is an epigenetic mark” (line 309), or “The study of Turan…to use of MEIS1 direct inhibitors” (line 431). I only have some concerns regarding the description and analysis of the MEIS1::GFP experiments. Indeed, from the images that are shown Figure 2, it appears that a substantial amount of fusion protein can enter the nucleus, although the authors claim that none of the MEIS1 mutant are able to enter the nucleus. I agree, that the nuclear fluorescence seems lower than the cytoplasmic one but still fluorescence can be clearly seen in the nucleus. The high difference in the histograms seems not to faithfully reflect what is seen in the images and I wonder whether running a quantification excluding the nucleoli would not be preferable and more precisely reflect actual differences between the cytoplasm and the nucleoplasm. At least, the authors should change their statement.

Author Response

Please find the letter in attachment 

Reviewer 2 Report

Summary

The manuscript entitled “MEIS1 in Haematopoiesis and Cancer. How Can Meis1-Pbx Interaction be Used in Therapy” by Francesco Blasi and Chiara Bruckmann aims at summarizing the current knowledge on the role of the Meis1 in MLL-r leukemic and provide new perspective for druggable therapy. In more detail, the authors provide overviews of the druggable approaches used to target HOXA9 and Meis1 in MLL-r Leukemia, as well as the physiological roles of Meis1 (and Prep1) in embryonic and adult haematopoiesis.

They further propose Meis1 as an ideal target for Leukaemia that could be supported by more in depth characterization of the protein at the molecular level.

Finally, they provide results indicating that targeting the MEIS1/PBX1 interaction, the two well described cofactors of HOX proteins offers promising outlook for anti-cancer therapy in general, and potentially in MLL-r leukaemia.

General comment

Overall the manuscript is interesting, well described concerning the overview of the existing therapies and proposes interesting perspectives. However, the overall flow seems difficult to follow and lacks connections. In particular, concerning the role of the Meis1/Pbx complex, the similarity of the function of Meis1 and Prep1 in haematopoiesis that seems quite different from the other tissues. The information is spread over the manuscript making the overall  molecular mechanism and function of the different complexes in haematopoiesis difficult to get and to envision.

In addition, the results provided in the manuscript showing new interface of interaction between Pbx and Meis1 or Prep1 are promising but seem out of context as nothing is speculated before on the role of the 2 complexes in Leukaemia.

The last paragraph and proposal is very interesting (targeting the cytoplasmic Meis1 rather the nuclear one) and makes sense with the other druggable approaches described.

Overall, one feels that the manuscript would benefit from large reshaping, by removing the results or integrating it clearly in the overall flow, and increasing the clarity of the role of Meis1, Prep1 /Pbx (1, 3) complex in haematopoiesis (maybe with a figure?) and their potential role in leukaemia as complex.

Moreover, the HOX TFs are major partners of the complex as stated in the abstract and highly developed in the paragraph describing the existing druggable approaches. Yet, the manuscript does not provide information regarding HOX related complex in haematopoiesis. It would be important to stress this point and the potential impact of these drugs on the Hox trimeric complex.

Other comments

-Line 117-188: However, this does not happen and represents a still unsolved part of the physiology of these two transcription factors [44].

It would be interesting to propose theory/hypothesis on this. Is it a question of dose? What is the potential involvement of the HOX TF in this complex and its function in haematopoiesis?

-Line 151-152: Meis1 ablation in adult mice influences normal haematopoiesis but is relatively well tolerated; therefore, meis1 functional inactivation in therapy might represent a valuable therapeutic target

This seems to lack connection. Why Meis1 would be a promising target if it absence is tolerated for normal haematopoiesis? Is their other MEIS proteins that may rescue the phenotype? Is it only misregulated in leukaemia while not required for normal haematopoiesis function? The hypothesis could be clearly stated by the author to re-enforce the proposal.

-Line 157-159: The lack of phenotype in double heterozygous embryos (Prep1 +/- -Meis1 +/-) was unexpected [44]. Therefore, even if both Prep1 and Meis1 affect embryonic haematopoiesis, they must act through different molecular pathways.

This seems particularly intriguing. How about Pbx1 in this context? Is the Meis1/Pbx1 and Prep1/Pbx1 could have some redundant function, thereby compensating the heterozygous mutant prep1/meis1 ? It has been briefly presented for the Pbx1/Meis1 in Zebrafish for the embryonic development. The manuscript would gain visibility by stating a clear proposal concerning the potential function of the Meis1/Pbx1 versus Prep1/Pbx1 complex in haematopoiesis.

-Line 449-452: In addition, the specific switch-off of MEIS1 upon VTP50469 treatment in MLL-r leukemia has demonstrated for the first time that HOXA9 downregulation is dispensable for the regression of leukemia, and that MEIS1 represents  the real target.

This sentence seems somehow too strong. Meis1 seems in that case, an ideal target compared to HOXA9 but not a “real target”.

Appendix:

The combination of review and data sounds confusing, in particular as no clear connexion has been done or proposed between Meis1 and Pbx1 complex in adult haematopoiesis, neither leukaemia. Is it a potential reactivation of the embryonic developmental program ? This would be interesting to mention.

-Line 485-486: All MEIS1-GFP-tagged constructs were successfully expressed in MEFs as judged by flow-cytometry analysis (data not shown).

It is rather an important data to show

-Line 578-579: Remarkably, Pbx3 has been shown to be the most important form of PBX that interacts with MEIS1 in human MLL-r leukemias [31,132]

This is an important information that could have been discussed earlier in the manuscript, concerning the role of the Pbx/Meis complex in haematopoiesis, and the redundancy or specificity of the different Pbx proteins in complex

Author Response

Pleas find the letter in attachment
